# Modification of Pea Starch Digestibility through the Complexation with Gallic Acid via High-Pressure Homogenization

**DOI:** 10.3390/polym14132623

**Published:** 2022-06-28

**Authors:** Franciene Almeida Villanova, Amy Hui-Mei Lin

**Affiliations:** Singapore Institute of Food and Biotechnology Innovation, Agency for Science, Research and Technology (A*STAR), Singapore 117609, Singapore; francienevillanova@hotmail.com

**Keywords:** pea starch, starch digestibility, resistant starch, gallic acid, phenolic acid, high-pressure homogenization

## Abstract

Pea starch and some legume starches are the side streams of plant-based protein production. Structural modification toward moderate digestibility and desirable functionality is a way to increase the economic values of these side-stream starches. We applied an innovative and sustainable technique, high-pressure homogenization, to alter pea starch structure, which resulted in a high level of complexation with the small phenolic acid molecule, gallic acid, to alter starch digestibility. This study showed a great level of disruption of the compact starch structure represented by the decrease in gelatinization temperature, enthalpy change, and relative crystallinity. The addition of a high concentration (10%) of gallic acid contributed to a typical V-type X-ray diffractometry pattern. Data demonstrated a significant decrease (~23%) in the susceptibility to α-amylase and an increase in resistant starch (~13%). In addition, starch functionality was improved with a reduced retrogradation rate. Pea starch responded to the high-pressure homogenization process well. Compared with the rice and maize starch reported in the literature, pea starch required a reduced amount of gallic acid to form a high level of complexation with a significant delay in starch digestion.

## 1. Introduction

Legumes are a primary source of plant protein, and pea (*Pisum sativum*) is a preferred crop as it is allergen-free and is typically not genetically modified [1,2]. The massive production of pea proteins generates much pea starch, which has low thermal stability, low acid and shear resistance, and a high tendency to retrograde [3]. Research to develop high-value-added applications with pea starch is in high demand [1,4,5,6]. Pea starch has a relatively high amount of amylose, approximately 40% (*w*/*w*), depending on the variety [7]. With its rich amylose content, pea starch has the potential to be a suitable carrier of micronutrients (e.g., phenolic acids) through complexation with amylose molecules. Among plant-derived micronutrients, gallic acid (GA) stands out as a small, natural crosslinker with three hydroxyl groups, forming either inclusion complexes, non-inclusion complexes, or both with starch molecules [8]. GA is a safe compound with healthy benefits, such as antioxidant and enzyme inhibition activities.

Complexation can also alter the functionality of pea starch to suit various food applications, such as reducing the tendency of retrogradation, improving heat and shear stability, and slowing starch digestibility [9]. However, conventional amylose-based complexation processing usually requires heating to gelatinize the starch, and starch with a high quantity of amylose, at 70% or higher, is preferred. In addition, some micronutrients used for complexation are sensitive to heat and degrade during heat-based complexation processing. Furthermore, the cost of high-amylose starch makes those products less competitive on the market. Therefore, we propose using innovative and cost-effective processing techniques, such as high-pressure homogenization (HPH), to utilize side-stream ingredients such as pea starch from plant-sourced protein production.

HPH is a sustainable food processing technique with high pressure and high shear force, and it promotes the interaction between starch and other ingredients, such as lipids and phenolic acids [10,11,12]. HPH effectively fosters the formation of starch–phenolic acid complexes by decreasing the starch molecular size, which promotes aggregation and microparticle formation. In addition, the leached starch molecules, during HPH processing, have a high tendency to react with phenolic acids under a high-speed shear force [8].

This study proposes using HPH to modify pea starch digestibility through complexation with GA. Pea starch has a C-type polymorphic structure, which is different from maize and rice starch that has been previously studied. The peripheral region of pea starch granules has an A-type polymorphic crystalline structure, which has a looser structure and is gelatinized at a lower temperature than the B-type polymorphic crystalline structure in the center area of pea starch granules. With these unique structural features, we expected HPH to stimulate increased leaching from the peripheral region due to its easy degradation, promoting the complexation of leached starch molecules with GA via the high shear force. Compared with conventional heating (~one hour or longer), the short processing time with HPH (~12 min in this study) may prevent the heat degradation of phenolic acids. In this study, we modified pea starch digestibility through complexation with GA via HPH, and the modification via HPH processing was characterized.

## 2. Materials and Methods

### 2.1. Materials

Commercial pea starch was received from Roquette (LN 30; Lestrem, France), and as noted by the manufacturer, the native pea starch contained 0.1% of crude fat and 0.5% of protein. Gallic acid (3,4,5-trihydroxybenzoic acid, 97–102%) was from Sigma-Aldrich (St. Louis, MO, USA). D-Glucose Assay Kit was purchased from Megazyme International Ltd. (Wicklow, Ireland). Pepsin from porcine gastric mucosa (371 U/mg), pancreatin from porcine pancreas (8x USP), amyloglucosidase from *Aspergillus niger* (319 U/mL), invertase from *S. cerevisiae* (334 U/mg), and α-amylase from porcine pancreas (type VI-B, 11 U/mg) were from Sigma-Aldrich (St. Louis, MO, USA). All reagents and chemicals were of analytical grade and used without further purification. Enzyme activity was defined and determined by the manufacturers, and newly purchased enzymes were used in this study.

### 2.2. Methods

#### 2.2.1. Preparation of Pea Starch–Gallic Acid Complex

Pea starch slurries (10% *w*/*w*) were prepared using deionized water at approximately 27 ± 5 °C. In sequence, GA (5% and 10% weight based on pea starch) was added into the starch slurry with constant stirring for 3 min. Then, the mixture was homogenized via a high-pressure homogenizer (Panda PLUS 2000, GEA Niro Soavi S.p.A., Italy) at 180 MPa for four cycles, which was 3 min for each cycle and a total of 12 min. A control treatment without GA was prepared with the same procedure. Due to the smooth warming of the mixtures promoted by the high-pressure homogenization process, the mixtures were cooled to approximately 30 °C and then washed twice with 50% (*v*/*v*) ethanol solution. The suspension was centrifugated (2800× *g* for 5 min) to remove free (uncomplexed) GA. The precipitates were dehydrated in an oven at 37 °C for approximately 14 h. Dehydrated samples were ground into powder using a coffee grinder (EKM150, Rommelsbacher ElektroHausgeräte GmbH, Dinkelsbühl, Germany). Powders were passed through a 100-mesh screen before further analyses.

#### 2.2.2. Thermal Properties by DSC

The thermal properties of the pea starch–GA complexes and of the control treatment were determined by differential scanning calorimetry (DSC; model 214 Nevio; Netzsch-Gerätebau GmbH, Selb, Germany). Samples (6 mg) were weighted in aluminum pans, mixed with 18 μL of deionized water, and then equilibrated at ambient temperature for 24 h. The pans were scanned from 10 °C to 150 °C at 10 °C/min, and onset (To), peak (Tp), and conclusion (Tc) temperatures, enthalpy change (ΔH), and the range of gelatinization temperature (calculated by subtracting To from Tc) were recorded. The starch gelatinization degree (GD) was obtained following the equation: DSG (%) = (1 − ΔH_TS_/ΔH_NS_) × 100; where ΔH_TS_ is the enthalpy change of HPH-treated starches and ΔH_NS_ is the enthalpy change of native pea starch.

#### 2.2.3. Retrogradation Degree by DSC

To investigate the effects of GA on starch retrogradation, samples scanned in Section 2.2.2 were stored at 4 °C for seven days and then rescanned following the method of Jacobson and BeMiller (1998) [13]. The degree of retrogradation (DR%) was calculated following the equation: DR (%) = (ΔHr/ΔHg) × 100; where ΔHr refers to the enthalpy change of the retrograded sample after storage and ΔHg is the enthalpy change of gelatinized samples.

#### 2.2.4. Crystalline Patterns by X-ray Diffractometry

The native pea starch, HPH-treated pea starch and starch–GA complexes were equilibrated in a chamber with 100% relative humidity for 24 h before X-ray analysis. The X-ray diffractograms were obtained using a D8 Discover X-ray diffractometer (Bruker, Karlsruhe, Germany) equipped with an IμSCu microfocus sealed tube, with a step size of 0.02°, a target voltage of 50 kV, a current of 0.8 mA, and a scan time of 300 s, and the samples were scanned from 3° to 35°. The relative crystallinity (RC) of the starches was obtained by using the software Origin 8.0 (OriginLab Corp., Northampton, MA, USA), and using the following equation as described by Vanier et al. (2019): RC (%) = (Ac/(Ac + Aa)) × 100; where Ac is the crystalline area and Aa is the amorphous area on the X-ray diffractograms [14].

#### 2.2.5. Complex Index by Folin–Ciocalteu Analysis

The GA content in the complexes was estimated by the Folin–Ciocalteu procedure as described by Chi et al. (2019) [15] and Sanchez-Rangel et al. (2013) [16]. The pea starch–GA complex (10 mg) was weighed into a 50 mL Falcon tube and suspended in 10 mL of dimethyl sulfoxide. In the sequence, the mixture was stirred in a vortex for 2 min and then centrifuged at 1800× *g* for 10 min. An aliquot of the mixture (15 μL) was diluted with deionized water (240 μL) in a 96-well microplate, followed by adding the Folin–Ciocalteu reagent (0.25 N, 15 μL). The mixture was kept on the bench for 3 min, and then 30 μL of sodium carbonate (1N) was added into the mixture. The final mixtures were kept at proximately 27 °C for 90 min in the dark. The absorbance at 760 nm was recorded using a plate reader and compared against a GA standard curve. The complex index was defined as the percentage of the detected GA amount in treatments to the amount of GA added in the modification.

#### 2.2.6. α-Amylase Hydrolysis

Starch–GA samples (500 mg) were mixed with a 25 mL solution (pH 6.9), which contained calcium chloride (0.1 mM), glycerol (0.2 mg/mL), and sodium azide (0.2 mg/mL). The substrate mixtures were prewarmed at 37 °C for 5 min, and then 5 mL of α-amylase suspension (0.33 U/mg starch) was added into the mixture. The hydrolysis was conducted in a water bath controlled at 37 °C. An aliquot of 0.5 mL was taken after 0, 5, 10, 15, 20, 40, 60, 80, 100, and 120 min and heated in a boiling water bath for 10 min to inactivate enzyme activity. The concentration of reducing sugars released by α-amylase hydrolysis was determined by the miniaturized Somogyi–Nelson assay [17].

#### 2.2.7. In Vitro Starch Digestibility Assessment

To determine the rapidly digestible starch (RDS), slowly digestible starch (SDS), and resistant starch (RS) fractions, in vitro digestibility of pea starch–GA complexes was carried out following the method of Englyst and Cummings (1992) [18]. Pepsin (2.65 g) was suspended in 0.05 mol/L hydrochloric acid (120 mL) containing ethanol (1 mL) and guar gum (0.6 g) to obtain the enzyme working solution A. The enzyme working solution B was prepared by suspending 9 g of porcine pancreatin in 60 mL of 0.1 mol/L calcium chloride solution, followed by centrifugation at 9000× *g* for 10 min. Subsequently, amyloglucosidase (1.75 mL diluted into 2.25 mL of 0.1 mol/L calcium chloride) and invertase (18 mg dissolved into 2 mL of 0.1 mol/L calcium chloride) were added to working solution B.

For the gastric phase, the starch–GA complexes (0.5 g) were dispersed in 5 mL of sodium acetate buffer 0.05 mol/L (pH 5.2) and equilibrated at 37 °C for 5 min. Then, 10 mL of enzyme working solution A and three glass marbles were added to the samples, and the mixture was incubated in a shaking water bath (150 rpm) at 37 °C for 30 min. In sequence, to start the intestinal phase, 5 mL of enzyme solution B was added to the mixture, and it was further incubated at 37 °C for 120 min. Aliquots of the hydrolysates (1 mL) were collected at times of 0, 20, and 120 min of intestinal phase simulation and placed in a boiling water bath (10 min) to inactivate the enzymes. The samples were centrifuged at 10,800× *g* for 20 min, and glucose content present in the supernatant was analyzed using a GOPOD reagent. The released glucose after 20 and 120 min was labeled as G20 and G120, respectively, by which RDS, SDS, and RS fractions were calculated as described in the formulas below:RDS = G20 × 0.9/TS
SDS = (G120-G20) × 0.9/TS(1)
RS = [TS-RDS-SDS]/TS(2)

TS means the total starch (TS) content of the complexes used for digestibility measurement, equal to 0.5 g.

#### 2.2.8. Statistical Analysis

Data analysis was carried out by analysis of variance (ANOVA). Tukey’s test compared the means at the 5% level of significance using SAS^®^ software (Statistical Analysis System, Cary, NC, USA).

## 3. Results

### 3.1. Thermal Properties and Retrogradation of Pea Starch–GA Complex

The thermal properties of the pea starch, HPH-treated pea starch (HPH-PS), and HPH-treated pea starch–gallic acid complex (HPH-PS-GA) were examined with DSC, and the results are presented in Table 1. We observed a relatively low enthalpy change in native pea starch in this study. The pea starch used in this study was a by-product from pea protein isolation, which might not reserve the integrity of starch granules as the commercial starch isolation, and had a disorganized granular structure.

HPH-treated samples had a reduction in onset (To), peak (Tp), and conclusion (Tc) temperatures and in enthalpy change (ΔH) compared to native pea starch, indicating that HPH facilitated starch gelatinization. The addition of GA, both at 5% and 10%, did not significantly change gelatinization temperature, although both treatments had a slightly higher peak temperature (Tp) than HPH-treated starch without GA (Table 1). The presence of GA in both concentrations significantly decreased the enthalpy change (ΔH) (*p* < 0.05), indicating a lower thermostability of the starch structure. HPH-treated samples had a narrower gelatinization temperature range than native pea starch, suggesting a higher homogeneity of amylopectin crystals in the HPH-treated samples compared to the native pea starch. With the presence of GA, HPH-PS-GA had a higher gelatinization degree than HPH-PS.

After cold storage for seven days, HPH-PS and HPH-PS-GA-5% samples had a higher retrogradation percentage (95% and 99%, respectively) than untreated pea starch (84.7%; Figure 1). Samples made with 10% of GA had the lowest retrogradation rate (64.3%), 20% lower than the untreated pea starch.

### 3.2. Crystalline Patterns by X-ray Diffractometry

Figure 2 shows a typical C-type crystalline pattern in native pea starch with signature peaks at 5°, 15°, 17°, 18°, and 23°. The pattern of HPH-treated pea starch (without adding GA) shifted to a B-type pattern with peaks at 2θ of 15°, 17°, 22.4°, and 24°. With 10% of GA (HPH-PS-GA-10%), a peak at 19.8° was observed, suggesting a V-type inclusion complex formation.

Figure 2 also demonstrated data on the relative crystallinity, in which native pea starch had the highest degree of relative crystallinity (63.30%), followed by HPH-PS (62.48%), HPH-PS-GA-5% (27.27%), and HPH-PS-GA-10% (24.53%). The addition of GA significantly reduced the relative crystallinity by approximately 35% compared with the native pea starch, indicating that the crystalline was less organized with the presence of GA.

### 3.3. Complex Index

Figure 3 shows the results of GA content and complex index of samples. Samples processed with 10% GA had a high level of the complex index (*p* < 0.05).

### 3.4. In Vitro Starch Digestibility

Results of the α-amylase hydrolysis kinetic study showed HPH-PS-GA-10% reduced approximately 23% of the digestion rate compared with HPH-PS (Figure 4). The addition of GA (in both HPH-PS-GA-5% and HPH-PS-GA-10%) increased RS levels (Figure 5) when compared with HPH-PS (control sample). The highest concentration of GA (HPH-PS-GA-10%) promoted an increase of 13% in RS. This study did not observe significant changes in rapidly digestible starch and slowly digestible starch. The substantial reduction in the susceptibility to α-amylase (Figure 4) was not reflected in the glucose production by fungal glucoamylase (amyloglucosidase; Figure 5). The α-amylolysis might release some GA and inhibit fungal glucoamylase from generating glucose effectively. An advanced study in quantifying the free GA of α-amylase hydrolysates and examining enzyme activities would provide a disclosure.

## 4. Discussion

HPH modification with GA greatly disrupted the highly organized starch molecular structure. The structural disruption was caused by the mechanical change, including high pressure and high shear force, during HPH processing. The outcome of such structural disruption was the enhancement of hydration during heating [19], which promoted gelatinization and reduced crystallinity, as observed in this study. The interaction between GA and starch molecules further weakened the interactions between starch molecular chains, which decreased the enthalpy change. The acidic environment caused a relatively homogenous structure with a narrower temperature change [20,21], which was also observed in this study (Table 1).

HPH modification with GA improved pea starch functionality with a lower retrogradation rate. HPH alone increased the retrogradation rate from 85% to 99%, and such a significant change (*p* < 0.05) might be due to the molecular leaching promoted by HPH. However, the complexation with GA reduced retrogradation, and this desirable outcome was achieved only with a high concentration of GA. The bonding between GA and starch molecules may hinder the interaction between starch molecules and delay retrogradation.

HPH modification with a high GA concentration significantly decreased the susceptibility of pea starch to α-amylase. However, modification with a low concentration of GA did not have much influence on starch digestion. The difference between the two treatments was in the level of complexation, which interrupts the binding of α-amylase onto starch molecules [22]. The high GA modification formed a V-type complex, a mutual structural characteristic of resistant starch type 5. The α-amylase hydrolysis demonstrated a high level of indigestible residues, and an assessment based on Englyst et al. showed an increase in resistant starch fraction.

Conventionally, HPH has neither generated a high degree of complexation [10,15,23] nor a high amount of resistant starch as achieved in this study. Liu et al. (2019) reported a 1% complex index of rice processed via preheating gelatinization and HPH with 10% GA [15]. Chi et al. (2019) produced a near 13% complex index of rice via HPH with a high quantity of GA, 50% (*w*/*w*) [15]. Chi et al. (2017) also generated a near 14% complex index of maize starch via HPH with 40% GA [23]. In this study, we hypothesized that the unique C-type polymorphic structure of pea starch might generate more leaching molecules and small starch fragments for complexation during HPH processing. Data of this study supported the hypothesis and reached nearly 30% complexation with only 10% of GA. In terms of starch digestibility, rice starch modified with 29% GA via the HPH process only generated 30% resistant starch, whereas in this study, only 10% GA was needed to produce 35% resistant starch [10].

HPH was selected for this study to overcome a few disadvantages in conventional heat-based treatments, such as heat damage to phenolic compounds and higher energy demand for a long heating process. Findings of the change of digestibility and functionalities in this study suggest HPH is an effective method for the industry to utilize pea starch. Researchers may consider extending the study to other legume starches with similar granular structures.

## 5. Conclusions

In modern diets, many processed starchy foods are rapidly digested into glucose. Much research has demonstrated various modifications of starch digestibility. This study aimed to demonstrate an innovative technique and “clean-label” approach (e.g., without using synthetic chemicals) utilizing side-stream starch to alter starch digestibility. We believe that the selection of initial material (starch) is critical because each starch from different botanical sources responds to technology in different manners. Pea starch and many other legume starches that primarily originate from the massive production of plant-sourced proteins have a distinguished granular architecture that varies from commercial cereal (i.e., maize, rice) and tuber starch (i.e., potato, cassava). This study demonstrated a successful and effective complexation of pea starch with GA through an innovative and sustainable manufacturing technique, high-pressure homogenization. A V-type inclusion complex was formed, the susceptibility to α-amylase was greatly reduced, and the retrogradation rate was significantly hindered with a relatively low amount of gallic acid compared with the complexation of cereal starches. Utilizing the natural starch structure with a suitable technique is an effective way to produce desirable functionality.

## Figures and Tables

**Figure 1 polymers-14-02623-f001:**
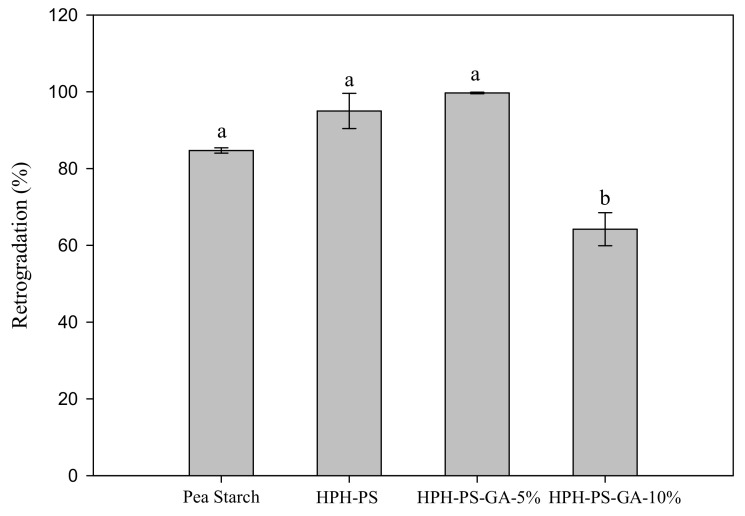
Retrogradation percentage of native and modified pea starch. HPH-PS: high-pressure homogenization-treated pea starch; HPH-PA-GA-5% and HPH-PA-GA-10%: high-pressure homogenization-treated pea starch with 5% or 10% gallic acid. The letters a and b indicate statistically significant difference at *p* < 0.05 between groups.

**Figure 2 polymers-14-02623-f002:**
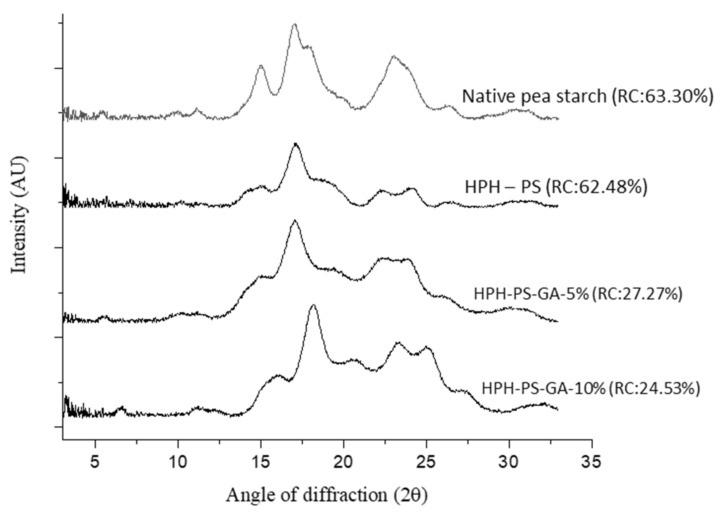
X-ray diffractograms of native and modified pea starch. HPH-PS: high-pressure homogenization-treated pea starch; HPH-PS-GA-5% and HPH-PS-GA-10%: high-pressure homogenization-treated pea starch with 5% and 10% gallic acid, respectively; RC: relative crystallinity.

**Figure 3 polymers-14-02623-f003:**
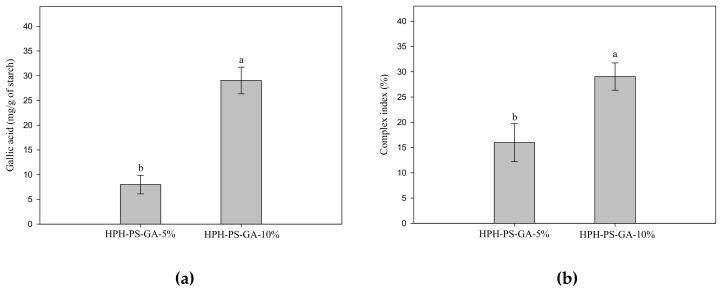
Gallic acid content (**a**) and complex index (**b**) of pea–gallic acid complexes made by high-pressure homogenization with 5% and 10% (*w*/*w*) gallic acid. The letters a and b indicate statistically significant difference at *p* < 0.05 between groups in the same graph.

**Figure 4 polymers-14-02623-f004:**
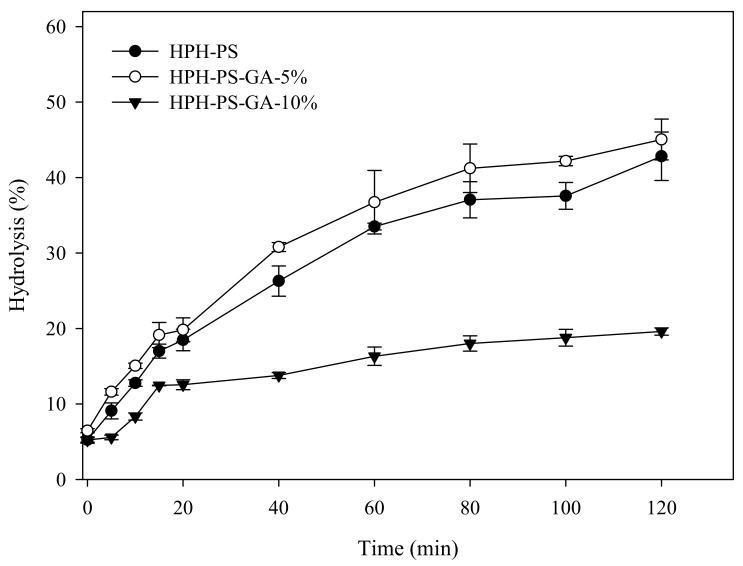
α-Amylase hydrolysis kinetics of high-pressure homogenization-processed pea starch with 0% (HPH-PS), 5% (HPH-PA-GA-5%), and 10% (HPH-PA-GA-10%) gallic acid.

**Figure 5 polymers-14-02623-f005:**
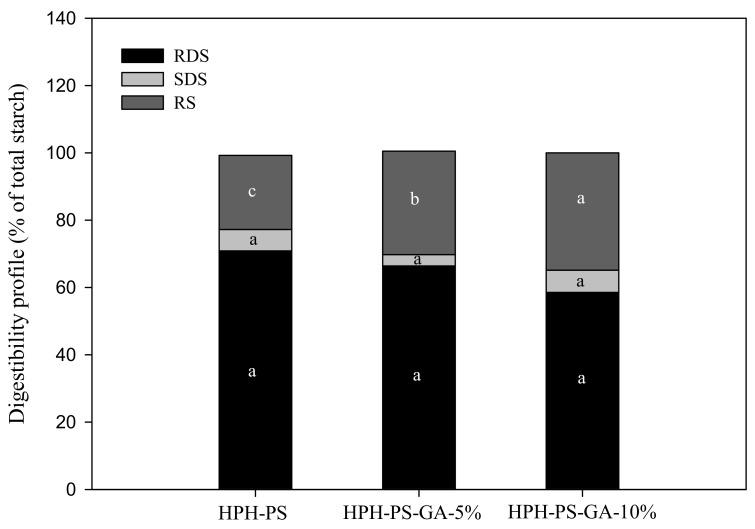
The in vitro starch digestibility of high-pressure homogenization-processed pea starch with 0% (HPH-PS), 5% (HPH-PA-GA-5%), and 10% (HPH-PA-GA-10%) gallic acid. Different letters in the same digestion category (i.e., RDS) indicate a significant difference at *p* < 0.05.

**Table 1 polymers-14-02623-t001:** Thermal properties and gelatinization degree of native and modified pea starch.

Treatment	To (°C)	Tp (°C)	Tc (°C)	ΔH (J/g)	ΔT (Tc-To)	GD (%)
Pea Starch	59.25 ± 0.45 ^a^	72.05 ± 0.15 ^a^	80.55 ± 0.05 ^a^	1.65 ± 0.14 ^a^	21.30	-
HPH-PS	49.70 ± 1.50 ^b^	60.40 ± 0.50 ^b^	69.95 ± 0.45 ^b^	0.77 ± 0.05 ^b^	20.25	53.69 ± 2.80 ^b^
HPH-PS-GA-5%	49.40 ± 0.80 ^b^	61.10 ± 0.80 ^b^	69.90 ± 0.80 ^b^	0.46 ± 0.06 ^c^	20.59	72.24 ± 3.89 ^a^
HPH-PS-GA-10%	51.10 ± 0.60 ^b^	62.30 ± 0.00 ^b^	68.55 ± 0.25 ^b^	0.35 ± 0.02 ^c^	17.45	78.81 ± 1.42 ^a^

Notes: Results are the M ± SD from duplicated analyses. Mean values in the same column with different letters are significantly different (*p* ≤ 0.05). Abbreviations: GD: gelatinization degree; HPH-PS: high-pressure homogenization-treated pea starch; HPH-PA-GA-5% and HPH-PA-GA-10%: high-pressure homogenization-treated pea starch with 5% or 10% gallic acid.

## Data Availability

Not applicable.

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
