# Peer review of "Modification of Pea Starch Digestibility through the Complexation with Gallic Acid via High-Pressure Homogenization"

_polymers, 2022, doi:10.3390/polym14132623_

Round 1

Reviewer 1 Report

line 52-58 "Gallic acid (GA) is a small natural crosslinker ..." - is an insertion in the purpose of the work. In my opinion, this is important information, but it should appear earlier in the introduction and not for research purposes.

line 85 "cooled to approximately 30 ° C ..." I don't understand and when it was warmed up, the mentioned temperature in the initial part of exactly this methodology was 27 degrees Celsius?

line 90 "pore size: 100 mesh" - the pores are present in the starch but not in the sieve. some of the pores may be closed on one side, which would not allow the screening process. I suggest removing the phrase pore size and leaving only 100 mesh in the parentheses

2.2.4. Complex index and 2.2.6. In vitro starch digestibility assessment - most of the methodologies discussed so far contain very detailed descriptions of their implementation - so I do not understand why this part basically contains names and not the actual method. a description is necessary so that the potential reader can repeat the experience.

3.1. Thermal properties of pea starch-GA complex - such a title indicates the discussion of the results from DSC, so what is the purpose of the discussion of the retrograde results here? it is a process, not a property, and completely different mechanisms !!!!!.

Author Response

Dear Reviewer,

Thank you for the feedback on our manuscript entitled “Modification of pea starch digestibility through the complexation with gallic acid via high-pressure homogenization.”. We revised the manuscript accordingly. Please also view the point-by-point responses below:

  1. Line 52-58 "Gallic acid (GA) is a small natural crosslinker ..." - is an insertion in the purpose of the work. In my opinion, this is important information, but it should appear earlier in the introduction and not for research purposes.

[Authors]: We edited more details about gallic acid in the revision.

  1. Line 85 "cooled to approximately 30 ° C ..." I don't understand and when it was warmed up, the mentioned temperature in the initial part of exactly this methodology was 27 degrees Celsius?

[Authors]: The complexes were prepared at room temperature (27 ± 5°C), but the high-pressure homogenization generated heat and increased the temperature to near 50 °C in our study. A similar phenome was also observed by Che et al. (2007). We had the slurries cool down to near 30 °C before proceeding with the described steps. A sentence was added in line 89 in the revision to explain the cooling step.

Reference mentioned: Che, L., Li, D., Wang, L., Özkan, N., Chen, X. D., & Mao, Z. (2007). Effect of High-Pressure Homogenization on the Structure of Cassava Starch. International Journal of Food Properties, 10(4), 911–922. doi:10.1080/10942910701223315

  1. line 90 "pore size: 100 mesh" - the pores are present in the starch but not in the sieve. some of the pores may be closed on one side, which would not allow the screening process. I suggest removing the phrase pore size and leaving only 100 mesh in the parentheses

[Authors]: The pore size refers to the size of the opening of the sieve screen. We revised the sentence.

  1. 2.4. Complex index and 2.2.6. In vitro starch digestibility assessment - most of the methodologies discussed so far contain very detailed descriptions of their implementation - so I do not understand why this part basically contains names and not the actual method. a description is necessary so that the potential reader can repeat the experience.

[Authors]: The description of the methods was provided in detail for both sessions 2.2.4. (Complex index) and 2.2.6. (In vitro starch digestibility assessment), as requested.

  1. 1. Thermal properties of pea starch-GA complex - such a title indicates the discussion of the results from DSC, so what is the purpose of the discussion of the retrograde results here? it is a process, not a property, and completely different mechanisms !!!!!.

[Authors]: The title of the sessions 2.2.2 and 3.1 was corrected as suggested. The retrogradation data of starch-GA complex was also obtained by DSC, where the scanned samples used to determine the thermal properties of starch-GA complexes were kept at 4 °C for 7 days and then re-scanned to analyze enthalpy changes. Thus, we compiled the results in the same session since the DSC technique can be used to analyze both variables (thermal properties and retrogradation process).

Reviewer 2 Report

My decision to this work is that it can be accepted with the following corrections that I state below that will be answered and corrected: 

The study presents the effect of high-pressure homogenizer to produce amylose-gallic acid complex. 

The pea starch shows low enthalpy value that is not common in native starches (10-15 J/g). This value shows that disorganization of starch components is present before the complex formation. This disorganization can be favoring the complex formation. 

Figure 1: Addition of GA retard starch retrogradation; however, the retrogradation level was similar to the HPH-PS sample. Why? 

DSC-traces should be including. 

Why the phase transition of the complex was not present? 

Figure 4: It is not clear how no difference between HPH-PS and HPH-PS-GA was found? 

The results of figure 4 does not agree with the figure 5. 

Author Response

Dear Reviewer,

Thank you for the feedback on our manuscript entitled “Modification of pea starch digestibility through the complexation with gallic acid via high-pressure homogenization.”. We revised the manuscript accordingly. Please also view the point-by-point responses below:

  1. An interesting article about the complexation of pea starch structure with gallic acid via high-pressure homogenization for altering starch digestibility. The gelatinization temperature, enthalpy change, and relative crystallinity of the complex decreased as a result of a great level of disruption of the compact polysaccharide structure due to its complexation with galic acid. Furthermore, pea starch required a reduced amount of gallic acid to form a high level of complexation compared to the rice and maize starch. DSC and XRD were the methods used for the characterization of the pea starch-gallic acid complex. Complex index and in vitro starch digestibility were also determined. The obtained results were interpreted from statistical point of view. The experimental data obtained are well explained and compared with other data known from the literature. Also, the references contain up-to-date literature. However, the conclusions must be rewritten, emphasizing the experimental results obtained after complexation of pea starch with gallic acid.

[Authors]: We appreciate the compliments about our study and thank the reviewer for the suggestion. We revised the conclusion section with an additional summary. We also take the opportunity to express our opinion about starch modification. Much research applies general techniques to modify starch without considering that each starch is unique in its structure. Pea starch responds to high-pressure homogenization much better than cereal starch because of the advantage of its granular architecture.

Revised conclusion:

  1. Conclusions

In modern diets, many processed starchy foods are rapidly digested into glucose. Much research has demonstrated various modifications of starch digestibility. This study aimed to demonstrate an innovative technique and “clean-label” approach (e.g., without using synthetic chemicals) utilizing side-stream starch to alter starch digestibility. We believe that the selection of initial material (starch) is critical because each starch from different botanical sources responds to technology in different manners. Pea starch and many other legume starches that primarily originate from the massive production of plant-sourced proteins have a distinguished granular architecture that varies from commercial cereal (i.e., maize, rice) and tuber starch (i.e., potato, cassava). This study demonstrated a successful and effective complexation of pea starch with gallic acid through an innovative and sustainable manufacturing technique, high-pressure homogenization. A V-type inclusion complex was formed, the susceptibility to α-amylase was greatly reduced, and the retrogradation rate was significantly hindered with a relatively low amount of gallic acid compared with the complexation of cereal starches. Utilizing the natural starch structure with a suitable technique is an effective way to produce desirable functionality.

Reviewer 3 Report

An interesting article about the complexation of pea starch structure with gallic acid via high-pressure homogenization for altering starch digestibility. The gelatinization temperature, enthalpy change, and relative crystallinity of the complex decreased as a result of a great level of disruption of the compact polysaccharide structure due to its complexation with galic acid. Furthermore, pea starch required a reduced amount of gallic acid to form a high level of complexation compared to the rice and maize starch. DSC and XRD were the methods used for the characterization of the pea starch-gallic acid complex. Complex index and in vitro starch digestibility were also determined. The obtained results were interpreted from statistical point of view.

     The experimental data obtained are well explained and compared with other data known from the literature. Also, the references contain up-to-date literature.

     However, the conclusions must be rewritten, emphasizing the experimental results obtained after complexation of pea starch with gallic acid.
